POINT OF VIEW

# Is cell size a spandrel?

**Abstract** All organisms control the size of their cells. We focus here on the question of size regulation in bacteria, and suggest that the quantitative laws governing cell size and its dependence on growth rate may arise as byproducts of a regulatory mechanism which evolved to support multiple DNA replication forks. In particular, we show that the increase of bacterial cell size during Lenski's long-term evolution experiments is a natural outcome of this proposal. This suggests that, in the context of evolution, cell size may be a 'spandrel'

**ARIEL AMIR**[*]

*For correspondence: arielamir@seas.harvard.edu

The question of how cells control their sizes has a long history (*Haldane, 1926*). The observation that cells of a given type often have a narrow size distribution suggests that cell size is tightly controlled, and one can try to rationalize this observation in various ways (*Young, 2006*; *Ginzberg et al., 2015*). For instance, one may speculate that diffusion will limit the supply of nutrients in large cells due to their small surface-area-to-volume ratio: however, many microbes are order of magnitudes smaller than any limit imposed by diffusion (*Landy, 2014*). Here I discuss the control of cell size in bacteria, which has been shown to manifest several size-related quantitative laws.

Cell size can be measured accurately, both on a population level and at the single-cell level. For several bacterial species it has been reported that the cell volume depends on its growth rate via a quantitative dependence known as the Schaechter-Maaløe–Kjeldgaard (SMK) growth law (*Schaechter et al., 1958*). For example, in *Escherichia coli* and its close relative *Salmonella typhimurium*, it is known that $V \propto e^{\lambda T}$, where $V$ is the cell volume, $\lambda$ is the cell growth rate, and $T$ is constant and equals 60 minutes at 37°C. (The growth rate is defined as $\lambda = log(2)/t_d$, where $t_d$ is the time taken for the number of cells in the sample to double).

Discovered in the 1950s, the SMK growth law is often interpreted in terms of cells adapting to their conditions: cells are larger at faster growth rates to accommodate more genetic material (*Wang and Levin, 2009*; *Chien et al., 2012*) or

more ribosomes (*Valgepea et al., 2013*). Here we will refute this interpretation and suggest that, in certain cases, size may be a 'spandrel' in the sense coined by Gould and Lewontin (*Gould and Lewontin, 1979*): in other words, cell size may be a "phenotypic characteristic that is a byproduct of the evolution of some other characteristic, rather than a direct product of adaptive selection" (to use the definition given by Wikipedia). The dependence of cell size on growth conditions, we shall argue, is not an adaptation but a causal consequence of a particular regulation scheme, whose primary purpose is otherwise.

We will outline a particular model of regulation in which the control of cell size occurs at the initiation of DNA replication. Within this model, which appears to capture many experimental results, a new round of DNA replication is initiated when the cell has *accumulated* a critical biomass (or volume) per each origin of replication from the previous initiation. This 'adder per origin' model has the virtue of tightly regulating the number of replication forks, and we will suggest that this should have more significant implications on cell fitness than changes in cellular dimensions. We also briefly discuss the potential molecular mechanisms for implementing this control strategy.

## Multiple replication forks and their regulation

In rich media, bacteria such as *E. coli* and *Bacillus subtilis* may have a doubling time shorter

than the duration of DNA replication. This remarkable feat is achieved by having several ongoing rounds of DNA replication, each initiating from the single *oriC* locus on the chromosome, most of which will only terminate in a future generation (*Yoshikawa and Sueoka, 1963*; *Yoshikawa et al., 1964*). In contrast to eukaryotes, bacteria initiate new rounds of DNA replication in a synchronous fashion: that is, the number of origins of replications double within a time frame that is short compared to the cell cycle duration (*Skarstad et al., 1986*). This implies that the number of origins on the circular chromosome will be of the form $2^n$ with n an integer (see *Zaritsky et al., 2011*) for a schematric illustration of the chromosome in fast growth conditions, and Figure 6 of *Hill et al., 2012* for examples of the origin number distributions at different growth conditions).

In a seminal result, Helmstetter and Cooper showed that over quite a broad range of doubling times (between 20 and 60 minutes), the time from the initiation of DNA replication to division is approximately $T = 60$ minutes (*Helmstetter et al., 1968*; *Cooper and Helmstetter, 1968*) (note that this is not true for slow growth conditions (*Adiciptaningrum et al., 2015*; *Wallden et al., 2016*), which we do not discuss here). This result was obtained using the 'baby machine' (*Helmstetter and Cummings, 1964*; *Helmstetter, 2015*), a clever experimental technique which approximately synchronizes the bacterial population. It may seem like a strange coincidence for this number to be identical to that appearing in the SMK growth law. Indeed, shortly after this discovery, Donachie published an elegant model which links these two observations (*Donachie, 1968*).

Donachie's main insight was to infer what the average cell size at the initiation of DNA replication is, based on the known value for the cell size at division (dictated by the SMK growth law). In order to make this extrapolation, additional information is necessary. First, what is the dependence of cell volume on time, at the single-cell level? Donachie assumed that cells grow exponentially over time. Later works verified that protein synthesis increases exponentially with time (*Cooper, 1988*), and more recently it was directly verified that the buoyant mass of cells also increases exponentially with time (*Godin et al., 2010*). These observations are consistent with the notion that the composition of the cytoplasm is approximately constant during the cell cycle. Taken together with the *constancy* of the length of time between initiation of

DNA replication and cell division, these results provide sufficient information to calculate the cell size at DNA replication initiation. The fact that $T = 60$ minutes appears in both the SMK growth law and in the context of DNA replication led Donachie to an intriguing conclusion: at the moment of initiation of DNA replication, *the average cell size is proportional to the number of origins per cell in that growth condition*. This has since been observed in other bacterial species (*Sharpe et al., 1998*), and has recently been directly verified in experiments where cell volume was measured at the time of DNA replication initiation (*Wallden et al., 2016*). Note that all of these experiments were done at relatively fast growth rates, though, and the results do not seem to hold for slow growth.

All this may naturally lead one to envision a mechanistic model, at the single-cell level. Cells can 'measure' their size, and initiate DNA replication at a critical size proportional to the number of origins in the cell. A constant time $T$ later, the cell divides. Note that in fast growing conditions, there would have already been several additional divisions between the initiation event and the consequent division events a time $T$ later. Hence this might correspond to division of the daughter or grand-daughter cell, rather than the same generation when initiation occurred. This model couples DNA replication, growth and division robustly, while elucidating the SMK growth law, which would be a causal consequence of the model. Nonetheless, we will suggest that there is a logical flaw in going between the population-level experiments which inspired the model, and the single-cell level proposed mechanism, and that it is inconsistent with additional experiments.

## Challenging Donachie's model

Recently, several studies have revealed an inconsistency between Donachie's model and experimental data, building on single-cell level data (*Amir, 2014*; *Osella et al., 2014*; *Campos et al., 2014*; *Taheri-Araghi et al., 2015*; *Soifer et al., 2016*). These experiments rely on the study of *correlations*. For instance, one may consider the Pearson correlation coefficient between mother and daughter cell sizes at birth (*Amir, 2014*); this coefficient is defined as the average of the product of the fluctuations of two variables around their means, normalized by the product of their standard deviations. It is 1 for perfectly correlated variables, $-1$ for perfectly anti-correlated variables, and 0 for independent

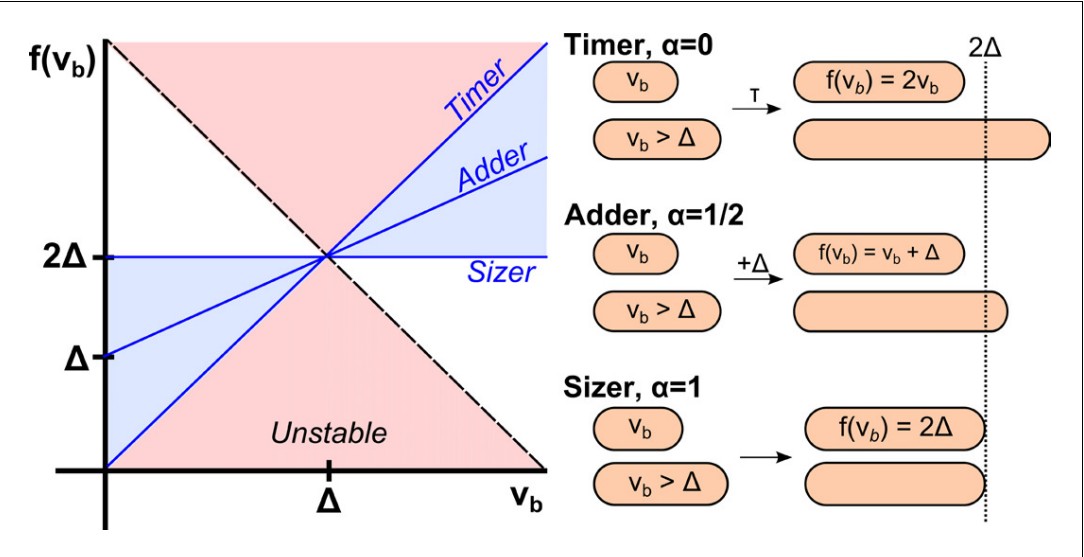

**Figure 1.** Phenomenological models for cell size control in bacteria. This graph shows how the size at division, $f(v_b)$, depends on the size at birth (x-axis) in three different models. The parameter $\alpha$ is related to the slope of the function, and can continuously interpolate across different models. Its value can be determined from single-cell level correlations (*Amir, 2014*; see *Marantan and Amir, 2016* for a recent generalization of this phase diagram). As $\alpha$ increases from 0 to 1, the correlations between mother and daughter cell sizes become weaker, yet the size distribution becomes narrower. The prevailing model for a critical size at initiation is effectively a 'sizer', and is inconsistent with recent experimental data supporting an 'adder'.

variables. Assuming a constant growth rate (which is a good approximation in many cases, including *E. coli* in fast growth conditions), it can be shown that a critical cell size for division will eliminate any correlations between cell size at birth and division (*Amir, 2014*). This is illustrated schematically in *Figure 1*. More generally, a critical cell size for entering a new cell cycle stage which precedes division by a constant time (as in Donachie's model) will also lead to vanishing correlations between birth and division size, i.e., a cell that is born smaller than average will take longer to reach that threshold size, but there will be no systematic bias for it to divide smaller than average – the memory of the initial conditions is 'washed away' by thresholding.

This prediction is in sharp contrast to numerous experimental findings on *E. coli* (*Koppes et al., 1980*; *Osella et al., 2014*; *Campos et al., 2014*; *Taheri-Araghi et al., 2015*; *Soifer et al., 2016*), where such correlations were measured and found to be significant and reproducible. In fact, these suggested that a cell that was born smaller than average by a volume $\Delta V$, will also be smaller than average *by the same amount* at division: in other words, when pooling all cells of a given size at birth together, their average size at division is related to that at

birth by the quantitative law $V_b = V_d + \Delta$. This law may, once again, be interpreted at the single-cell level as implying that a cell 'attempts' to add a constant volume from birth to division: that is, it implies that the cue for division is the accumulation of sufficient volume from cell birth. This scenario is referred to as the incremental or adder model. Recent work proposed a similar model, in which a constant surface area is added between birth and division (*Harris and Theriot, 2016*). Clearly, these models are different than Donachie's – yet they both stem from empirical findings. How can we reconcile Donachie's analysis with the observed correlations?

There is a simple way to resolve this apparent contradiction. The key point is to get away from the 'birth-centric' picture, where the cell is attempting to add the constant volume increment from birth to division, and replace it with a model where the volume is added between two DNA replication initiation events. Such a model recovers identical correlations between the various cell cycle events (*Ho and Amir, 2015*), while at the same time reproducing the exponential dependence of size on growth-rate, and hence is consistent with Donachie's observations.

## Implementing size control at DNA replication initiation

What does it mean to add a volume between two DNA replication initiation events? *Sompayrac and Maaloe, 1973* provide a hypothetical model where an 'initiator' protein is accumulated as the cell volume grows, and where initiation of DNA replication occurs when sufficient copies of that initiator are accumulated, thereby providing a proof-of-principle that a cell may indeed add a constant volume increment without measuring its absolute size at any point in time. Note that this model assumes that the accumulation of the initiator does not depend on the status of the chromosome, but only on the increase in volume: this is accomplished by a mechanism involving two genes on the same operon. The first encodes an autorepressing protein, whose concentration will thus be fixed and independent of growth rate (or gene dosage effects), while the second is the initiator, whose copy number reflects the relative increase in volume.

In fact, there are two variants of this model which we may consider: in the first, the initiator protein localizes at some point within the cell, and initiation of DNA replication occurs when its copy number reaches some threshold. This implies that the cell will add the same volume regardless of the number of replication forks (*Campos et al., 2014*). In a related model, the initiator is localized at the (potentially numerous) origins of replication (or some locus close to it), and hence for conditions of multiple replication forks the cell has to accumulate *more* volume – in proportion to the number of origins of replication.

Despite their deceiving similarity, these models are very different from each other, and their differences are crucial with regards to the question of cell cycle regulation. The first model does not lead to a narrow size distribution – in fact, it does not even lead to a stable size distribution, as was first noted in 2014 (*Campos et al., 2014*)! The reason for this is that when the noise occasionally leads to a cell cycle where no initiations occur – or two initiations occur – there is no feedback in the mechanism to allow the cell to recover the correct number of origins.

However, the second model (*Ho and Amir, 2015*), which we refer to as the 'adder per origin' model, leads to stable size distributions, and appears to explain all experimental findings discussed above. Within this model, division still occurs a constant time $T$ after initiation of DNA replication – as it does in the Helmstetter-Cooper model – but the trigger for initiation of a new round of DNA replication is accumulation of a constant volume *per origin of replication*. This seemingly innocuous change of the model of *Campos et al., 2014* is what provides the necessary feedback to recover from a faulty cell cycle not having a single initiation event. One should be cautious regarding interpreting this model as necessarily involving the *accumulation* of an initiator protein. While this is a possible mechanism, it is not the only way to implement this phenomenological model; in fact, it can be shown that other biophysical models based on *inhibition* rather than accumulation of a protein are mathematically equivalent and can also implement the incremental model (*Soifer et al., 2016*), thus the phenomenological observation of the 'adder per origin' model cannot unveil the structure of the particular molecular network that operates *in vivo*. Interestingly, it is the inhibition-based model that was first proposed (*Pritchard et al., 1969*), and provided the inspiration for the accumulation-based model.

## Is size the driver or passenger?

The adder per origin model has another appealing feature – it regulates the *number* of multiple replication forks. If the cell has the wrong number of origins of replication for the given growth condition, the number of replication forks will automatically adjust until the appropriate number is achieved. If a cell has too many origins of replication, the accumulation of the initiator at each origin will slow down (since the volume increment is effectively divided between all origins of replication), and hence the frequency of initiation will decrease until the number of origins per cell reaches the correct value. Note that all of the control occurs at the level of DNA replication initiation, with division occurring deterministically a time $T$ later – regardless of the cell size – in line with recent experimental findings (*Wallden et al., 2016*).

Within this model, both the number of origins and cell size scale exponentially with the growth rate, and hence are proportional to each other. Yet size and the number of replication forks play a very different role in the bacterial cell cycle: in steady-state growth bacteria *must* have an exponential dependence of origin number on growth rate (*Bremer and Churchward, 1977*; *Ho and Amir, 2015*), in order to guarantee that DNA replication will not become a bottleneck for cell

proliferation – which is why multiple replication forks presumably evolved in the first place. (Note that although the number of origins is an integer power of 2 in every cell, the aforementioned exponential dependence is a result of averaging over the entire population and thus is not restricted to take integer values). Changing this dependence will no doubt have significant consequences on fitness. Yet changes in cell size do not have that effect, and we may change cell size by tens or hundreds of percent without any measurable effect on the doubling time (which can typically be determined to accuracy within a few percent). For instance, thymine-limitation slows down the replication rate: this extends the duration of DNA replication, which is known as the C period, and leads to significantly larger cells with unperturbed doubling times (*Pritchard and Zaritsky, 1970*; *Zaritsky and Pritchard, 1973*; *Zaritsky et al., 2011*). Similarly, in ftsZ mutants the time between termination of replication and cell division (known as the D period) is changed: this leads to changes in the cell volume but does not change the doubling time (*Palacios et al., 1996*; *Hill et al., 2012*; *Zheng et al., 2016*). This raises the possibility that this control mechanism evolved in the first place not in order to tightly control size but, rather, as a means to control the number of multiple replication forks. Within this model, the exponential dependence of cell size on growth rate is not an adaptation of any sort but, rather, is a causal consequence of this particular regulation scheme.

What consequences does this interpretation lead to? It is known that many mutations affect cell size (*Chien et al., 2012*; *Yao et al., 2012*). Often, observing a large phenotypic response to a particular mutation is used as evidence supporting the role of a particular gene in the regulatory pathway. This logic may be problematic, however. Within the model's framework, *any* mutation that would affect the growth rate $\lambda$ (say, by affecting metabolism or the ribosome efficiency) or the duration of the $C$ and $D$ periods (thus affecting $T = C + D$), would affect cell size. Moreover, the effect can be quantified, since cell size should be proportional to $e^{\lambda T}$. Comparing this prediction with experimental data on various mutants shows quantitative agreement (*Ho and Amir, 2015*), as well as experiments where $T$ is systematically perturbed by controlling the expression level of proteins associated with cell division (*Zheng et al., 2016*).

A different class of experiments where this idea can be tested are the laboratory evolution experiments pioneered by Lenski and co-workers, in which bacteria are grown over thousands of generations, and their fitness is observed to continuously increase over time (*Lenski and Travisano, 1994*; *Lenski and Mongold, 2000*; *Lenski, 2004*; *Wiser et al., 2013*). In these experiments fitness should be strongly correlated with the growth rate, and inversely proportional to the doubling time (this would be precise in the case where cell growth is in exponential phase and is not affected by growth of other cells in the culture). It is observed that together with the fitness increase, cell size also increases, see *Figure 2*. According to our model, this is not an adaptation of the cell to

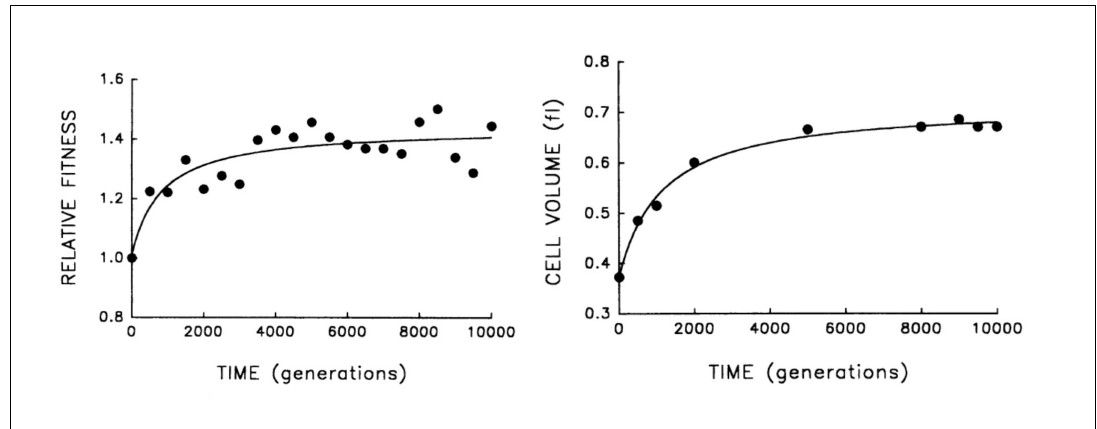

**Figure 2.** Bacterial evolution and growth regulation. *E. coli* cells evolving in a culture tube show both increasing fitness over time as well as increasing size. This can be naturally explained by a simple, specific regulation strategy, consistent with additional experiments. Figure adapted from *Lenski and Travisano, 1994*, with permission.

the growth conditions, but a causal consequence of the increase in growth rate, which is presumably what is being selected for. Note that we expect the growth rate to be under strong selection, and not $T$. While the latter affects the number of DNA replication forks, it should not have much impact on the growth rate – since the essence of multiple replication forks is precisely in bypassing the constraint imposed by DNA replication on the cell's doubling time. It would be interesting to study the dynamics of the duration $T$ in these evolution experiments, which previous results suggest might even increase over time (*Mongold and Lenski, 1996*), enhancing further the effect on cell size.

## Discussion

Although it is intuitive to put bacterial cell size center stage in light of the various quantitative laws which it has been shown to follow, we suggest here a different view, that the main constraints on the cell cycle architecture, in fast growing bacteria, stem from the necessity of correctly controlling the number of multiple replication forks. We discussed a model in which control occurs over the initiation of DNA replication – rather than cell division. The model is able to reconcile observations regarding the bacterial cell size which *a priori* seem to be contradictory. Importantly, the very same model regulates not only size, but also the number of replication forks – allowing the cell to efficiently proliferate in differing environmental conditions.

Cell size correlations similar to those seen in bacteria have also been observed in the budding yeast *Saccharomyces cerevisiae* (*Soifer et al., 2016*). In the case of budding yeast, recent work enhances our molecular-level understanding of cell size control (*Schmoller et al., 2015*), and some progress was made relating this molecular mechanism to phenomenological models (*Soifer et al., 2016*). It is also known that budding yeast cell volume is dependent on the growth rate (*Tyson et al., 1979*). It would be interesting to explore whether this dependence could arise from cell cycle 'architecture' (since the term spandrel has its origins in architecture). The numerous differences between the eukaryotic and bacterial cell cycles (for instance, that initiation occurs from many origins on each chromosome, asynchronously) suggests that a different kind of spandrel – yet to be determined – might be needed to elucidate this observation. In any case, it seems that being cautious in our interpretation of such dependencies (that is,

interpreting cell size dependence on growth rate as cellular adaption) could potentially be a lesson more broadly applicable.

All of the above discussion hinged on phenomenological observations at the single-cell level, yet hardly any mention of particular molecular mechanisms was made. Ultimately, further work should be able to bridge the gap between our improving phenomenological understanding of the coupling between cell growth, division and DNA replication, and our understanding of the molecular mechanism underlying these processes. The two approaches, molecular level and phenomenological studies, are complementary, and may supplement and benefit each other. The various quantitative laws discovered in bacteria are, in many ways, microscopes that provide us with new insights into the cell cycle. What we see with them should also inform and enhance our understanding at other levels, putting major constraints on the potential molecular mechanisms.

## Acknowledgements
We thank Richard Lenski, Pankaj Mehta, Andrew Murray and Nancy Kleckner for useful discussions. We thank Arieh Zaritsky and the reviewers for careful reading of the manuscript and for their important comments, and Po-Yi Ho for help with *Figure 1*.

**Ariel Amir** is in the School of Engineering and Applied Sciences, Harvard University, Cambridge, United States

 http://orcid.org/0000-0003-2611-0139

*Competing interests:* The author declares that no competing interests exist.

## Funding
No external funding was received for this work.

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
