## [Decision Letter]

Thank you for submitting your article "Is cell size a spandrel?" to *eLife* for consideration as a Feature Article. Your article has been reviewed by three peer reviewers, and the evaluation has been overseen by Naama Barkai as the Reviewing and Senior Editor. One of the reviewers, Stephen Cooper (Reviewer 1), has revealed his identity.

The reviewers have discussed the reviews with one another and the Reviewing Editor has drafted this decision to help you prepare a revised submission.

Summary:

The manuscript puts forward the suggestion that the well-studied mechanisms that regulate cell size in bacteria did not evolve for size-regulation. Rather, size regulation is a 'by-product' of the need to regulate the number of on-going replication forks. This conclusion is based on recent advances in single-cell experiments and theoretical approaches, which have led to new understanding of the size control mechanism.

Main Suggestion:

As reviewer #3 noted, the differences between classical ideas and the new model is not emphasized clearly enough. This should be fixed. In the discussion between reviewers, it was suggested to add a simple drawing that shows the difference in birth volumes predicted by the two models ('old' model and 'new' one, please see main comment Reviewer #3) during what is referred to as a noisy or faulty cell cycle. For instance, one could show that if a cell happened to initiate at some time after it had reached the critical volume/origin and then produced larger-than-average progeny, the timing and volume increase for the subsequent division would differ in the two models, with the new model being consistent with the recent findings with single cells. This type of drawing is thought to better clarify the main thrust of this manuscript.

Reviewer #1:

At the outset I want to apologize for the unremitting negative tone of this review. There are some positive points in the submitted article, but for the most part I feel that this paper is wrong on so many issues that I will try to be as detailed as possible in why I find these proposals most problematic.

Perhaps the best way to begin is to state clearly what I believe is the currently correct view of how the bacterial cell controls its size and progression through the division cycle. I believe that initiation of DNA synthesis occurs when there is an amount of "initiation mass" present for each origin and at that time initiations of DNA replication take place. Throughout the cell cycle I believe that mass is made simply and exponentially with mass increasing in proportion to extant mass at each time during bacterial growth during the cell cycle. I also believe that there are no variations in the composition of the "cytoplasm" of the cell during the cell cycle and that mass grows simply and exponentially during the cell cycle. I do not know precisely how initiation occurs, but following initiation of DNA replication over a range of growth rates the rate of DNA replication is relatively invariant and the time after termination of a round of replication till cell division is also relatively invariant. As pointed out by Dr. Amir in his review (which in that part is quite good) the size variation of cells as a function of growth rate is simply determined by this invariance of DNA replication and time from termination to division. The data from 1958 and 1968 fit these ideas extremely well.

In contrast to this simple model I feel that many of the suggestions made by Dr. Amir are akin to the Ptolmaic system of Astronomy. Whenever there a deviation from the predictions of the Ptolmaic system arose, a new epicycle is added to the extant cycles and more and more epicycles are used to make the observations fit to the Ptolmaic system. Analogously, I find the papers cited in Dr. Amir's paper, and in the submitted paper, an attempt to take weak data based on single cell measurements which are questionable (in my viewpoint) to fit the data by adding more and more slight variations in order to accommodate some published data.

I must admit that I have tried to look up a number of the papers referred to in the submitted paper, and I have tried to analyze these deeply, but time constraints have made it impossible for me to write a complete analysis of the papers used in this review to support the proposals made by Dr. Amir-Specifically the "Adder" system-and that is what I find problematic about this paper.

…

Reviewer #2:

I think this piece is worth publishing as front matter to make a more people realize that bacteria do not really care about their size. This is important in the light of recent high profile papers with a focus on bacterial cell size. I mainly like the spandrel analogy, but otherwise the observation that cell-size is a passenger and not driver is clear already from Cooper-Helmstetter-Donachi and more recently Wallden et al.

Reviewer #3:

The fundamental idea presented in this manuscript adds an important element to our understanding of bacterial cell cycle regulation and cell size determination. It basically states that the timing of a chromosomal initiation-of-replication event is determined by processes that take place subsequent to the previous initiation event, possibly but not necessarily, accumulation of an "initiator". As a consequence, the model places cell size control in the hands of initiation, and demands new thinking as to how initiation, and as a consequence the cell cycle, are controlled. It is a very satisfying idea because it is consistent with decades of information on the relationship between replication initiation and cell size, and with recent findings that cells growing at a specific rate add, on average, a constant volume from birth to division.

My one concern is whether the very important distinction between the Amir's proposal and previous proposals has been presented forcefully enough for readers to grasp its importance, especially those who may not be intimately familiar with the literature in this field. For instance, the essence of the model, which forms the basis for this manuscript and the entire cell cycle/size control idea, isn't presented until the middle of subsection “Challenging Donachie's model”: "…the volume is added between two DNA replication initiation events." After that model statement, there are comments regarding his proposal such as "…accumulation of a constant volume per origin of replication", "…accumulation of an initiator protein.", and "…multiple origins accumulation model…". These latter statements, taken alone, read too much like the old ideas that were proposed almost 50 years ago and still repeated even in papers published this year, i.e., that initiation takes place whenever a fixed active unit of initiator has accumulated per chromosomal origin (which is reflected in a fixed size/origin), and then cells divide C+D min later. That old proposal, as satisfying as it has been all these years, is wrong if taken literally and not modified, as proposed by the author, to state that it matters when the accumulation takes place, in order to be consistent with current data on cell size at division. I simply suggest that Amir make some changes in the manuscript so that the novelty of his ideas and their significance stand out better. In my opinion, this should also include changing the description of his model from "multiple origins accumulation" to something that reflects when this accumulation takes place. Whether or not Amir's model is correct, it is the most interesting idea on the overall picture of cell cycle control to appear in some time and needs to be promoted as a new way to think about the molecular basis of the initiation process.

---

## [Author Response]

Main Suggestion:

As reviewer #3 noted, the differences between classical ideas and the new model is not emphasized clearly enough. This should be fixed. In the discussion between reviewers, it was suggested to add a simple drawing that shows the difference in birth volumes predicted by the two models ('old' model and 'new' one, please see main comment Reviewer #3) during what is referred to as a noisy or faulty cell cycle. For instance, one could show that if a cell happened to initiate at some time after it had reached the critical volume/origin and then produced larger-than-average progeny, the timing and volume increase for the subsequent division would differ in the two models, with the new model being consistent with the recent findings with single cells. This type of drawing is thought to better clarify the main thrust of this manuscript.

*Reviewer #1:*

At the outset I want to apologize for the unremitting negative tone of this review. There are some positive points in the submitted article, but for the most part I feel that this paper is wrong on so many issues that I will try to be as detailed as possible in why I find these proposals most problematic.

Perhaps the best way to begin is to state clearly what I believe is the currently correct view of how the bacterial cell controls its size and progression through the division cycle. I believe that initiation of DNA synthesis occurs when there is an amount of "initiation mass" present for each origin and at that time initiations of DNA replication take place. Throughout the cell cycle I believe that mass is made simply and exponentially with mass increasing in proportion to extant mass at each time during bacterial growth during the cell cycle. I also believe that there are no variations in the composition of the "cytoplasm" of the cell during the cell cycle and that mass grows simply and exponentially during the cell cycle. I do not know precisely how initiation occurs, but following initiation of DNA replication over a range of growth rates the rate of DNA replication is relatively invariant and the time after termination of a round of replication till cell division is also relatively invariant. As pointed out by Dr. Amir in his review (which in that part is quite good) the size variation of cells as a function of growth rate is simply determined by this invariance of DNA replication and time from termination to division. The data from 1958 and 1968 fit these ideas extremely well.

In contrast to this simple model I feel that many of the suggestions made by Dr. Amir are akin to the Ptolmaic system of Astronomy. Whenever there a deviation from the predictions of the Ptolmaic system arose, a new epicycle is added to the extant cycles and more and more epicycles are used to make the observations fit to the Ptolmaic system. Analogously, I find the papers cited in Dr. Amir's paper, and in the submitted paper, an attempt to take weak data based on single cell measurements which are questionable (in my viewpoint) to fit the data by adding more and more slight variations in order to accommodate some published data.

I must admit that I have tried to look up a number of the papers referred to in the submitted paper, and I have tried to analyze these deeply, but time constraints have made it impossible for me to write a complete analysis of the papers used in this review to support the proposals made by Dr. Amir-Specifically the "Adder" system-and that is what I find problematic about this paper.

I would like to thank Prof. Cooper for his thorough review. I have made the following changes to the manuscript to address his comments:

1) Replaced “Growth-law” with “Schaechter’s growth-law throughout”.

2) Added additional references.

3) Added definition of Pearson correlation coefficient, and Figure 1 to explain the different correlations associated with the different models.

Reviewer #2:

I think this piece is worth publishing as front matter to make a more people realize that bacteria do not really care about their size. This is important in the light of recent high profile papers with a focus on bacterial cell size. I mainly like the spandrel analogy, but otherwise the observation that cell-size is a passenger and not driver is clear already from Cooper-Helmstetter-Donachi and more recently Wallden et al.

I thank the reviewer for his positive review, and for the correction they suggested, which was implemented in the revised version.

Reviewer #3:

The fundamental idea presented in this manuscript adds an important element to our understanding of bacterial cell cycle regulation and cell size determination. It basically states that the timing of a chromosomal initiation-of-replication event is determined by processes that take place subsequent to the previous initiation event, possibly but not necessarily, accumulation of an "initiator". As a consequence, the model places cell size control in the hands of initiation, and demands new thinking as to how initiation, and as a consequence the cell cycle, are controlled. It is a very satisfying idea because it is consistent with decades of information on the relationship between replication initiation and cell size, and with recent findings that cells growing at a specific rate add, on average, a constant volume from birth to division.

*My one concern is whether the very important distinction between the Amir's proposal and previous proposals has been presented forcefully enough for readers to grasp its importance, especially those who may not be intimately familiar with the literature in this field. For instance, the essence of the model, which forms the basis for this manuscript and the entire cell cycle/size control idea, isn't presented until the middle of subsection “*Challenging Donachie's model”*: "…the volume is added between two DNA replication initiation events." After that model statement, there are comments regarding his proposal such as "…accumulation of a constant volume per origin of replication", "…accumulation of an initiator protein.", and "…multiple origins accumulation model…". These latter statements, taken alone, read too much like the old ideas that were proposed almost 50 years ago and still repeated even in papers published this year, i.e., that initiation takes place whenever a fixed active unit of initiator has accumulated per chromosomal origin (which is reflected in a fixed size/origin), and then cells divide C+D min later. That old proposal, as satisfying as it has been all these years, is wrong if taken literally and not modified, as proposed by the author, to state that it matters when the accumulation takes place, in order to be consistent with current data on cell size at division. I simply suggest that Amir make some changes in the manuscript so that the novelty of his ideas and their significance stand out better. In my opinion, this should also include changing the description of his model from "multiple origins accumulation" to something that reflects when this accumulation takes place. Whether or not Amir's model is correct, it is the most interesting idea on the overall picture of cell cycle control to appear in some time and needs to be promoted as a new way to think about the molecular basis of the initiation process.*

I would like to thank the reviewer for his thoughtful comments and useful suggestions.

In the revised version I try to make the distinction between the previous models and the recent ones clearer (and earlier in the paper), and added Figure 1 to emphasize this distinction. I also changed “Multiple origins accumulation” to the hopefully clearer “adder per origin”.